# Difference in Germination Traits between Congeneric Native and Exotic Species May Affect Invasion

**DOI:** 10.3390/plants13040478

**Published:** 2024-02-07

**Authors:** Julieta Salomé-Díaz, Jordan Golubov, Luis E. Eguiarte, Alberto Búrquez

**Affiliations:** 1Posgrado en Ciencias Biológicas, Unidad de Posgrado, Universidad Nacional Autónoma de México, Ciudad de México 04510, Mexico; biojulieta@gmail.com; 2Department of Evolutionary Ecology, Instituto de Ecología, Universidad Nacional Autónoma de México, Ciudad de México 04510, Mexico; fruns@unam.mx; 3Plant Taxonomy and Systematics Laboratory, Universidad Autónoma Metropolitana, Ciudad de México 04960, Mexico; 4Estación Regional Noroeste, Instituto de Ecología, Universidad Nacional Autónoma de México, Hermosillo 83250, Mexico; montijo@unam.mx

**Keywords:** *Eragrostis*, invasion potential, Mexico, Pedregal de San Angel Ecological Reserve, Poaceae, propagule pressure, seed bank

## Abstract

Germination traits are components of invasion potential, and comparing seed traits in sympatric native and invasive species can offer insights into the invasion process. We characterized seed germination traits and how they influenced the success of *Eragrostis mexicana*, a native species, and *Eragrostis tenuifolia,* an exotic species (Poaceae) in Mexico, in the context of their potential for biological invasion. Seeds from both species were collected from four sites in a natural protected area in Mexico City, and the germination of seeds of different ages was conducted in experiments at different temperatures. *E. tenuifolia* exhibited higher germination percentages than the native *E. mexicana* across all treatments. Seed age had differential effects, with older seeds of the native *E. mexicana* germinating better, while *E. tenuifolia* performed better with younger seeds. Temperature positively impacted germination for both species, although *E. mexicana* was limited at lower temperatures. Exotic *E. tenuifolia* can germinate over a wider temperature range with earlier germination rates, and generate a seed bank lasting several years, which may contribute to naturalization. The importance of germination traits in the context of invasive species establishment underscores the potential role of seed banks in facilitating biological invasions.

## 1. Introduction

The impact of invasive species (IAS) on ecosystems has generated interest in identifying traits enabling their success at each stage of the introduction–naturalization–invasion continuum [1,2,3,4,5]. Despite debates regarding the existence of unique characteristics that favor invasive species [6], various traits and processes identified at the seed–seedling stage enhance species’ invasion potential, including seed size, seed accessories, seed viability, germination rates and requirements, and seed bank persistence [7,8,9].

Grubb [10] stated that if conditions were only suitable for adult plant development but not for the first stages of population regeneration, plant species could not locally persist, much less invade a given location. Tolerance of environmental conditions during the crucial stages of dispersal, germination, soil persistence as propagules, and early survival is of paramount importance to the process of species invasion. Persistent seed banks contribute to geographic and temporal dispersal and increase the probability of naturalization [8,9]. By ensuring that germination is distributed over time [11] and providing a source of genetic variation [12], seed banks can support species in overcoming limitations imposed by population size and environmental variation. They may also serve as a source of genotypic variation in the post-introduction evolutionary process.

The presence of seed banks has been regarded as a component of propagule pressure, which refers to the size and number of introduction events for a species to a given area [13], one of the key components of successful biological invasions [9]. Seed bank formation depends on the time between dispersal and eventual germination. Seed bank dynamics are affected by seed latency, germination rates, predation, senescence, and seed rain inputs [14]. Germination rate and timing are decisive in the process of plant establishment and directly impact the fitness, survival, and distribution of species [15,16,17]. Moreover, they determine the phenological pattern of subsequent life stages [18,19] and can affect interspecific interactions in communities [20]. Seed traits that affect germination are, therefore, crucial in the early stages of biological invasion (e.g., transport and establishment). They can also significantly influence subsequent stages in the continuum, including naturalization and dispersal.

Broad environmental tolerance across various life stages has been linked to successful biological invasions [21,22,23]. With this premise, species that can germinate across a wide environmental range and have a flexible germination cue may be equipped for successful invasion [24]. Widely tolerant species thrive in changing or unstable environments by engaging in early germination, meaning that they can outcompete native species [25]. Rapid response to environmental conditions conducive to germination (from a latent to an active seed bank) is beneficial to the invasion process [26,27]. Some invasive species germinate faster than ecologically or phylogenetically related native species, even when comparing seeds of the same species from native and invasive ranges [28,29,30]. The timing of germination not only determines future ecological interactions but also the environmental conditions experienced by seedlings [11,15,16,31]. Lowering the degree of synchrony in germination provides an opportunity to diversify the environmental conditions that seedlings experience, thereby increasing the probability of survival [32,33] and providing a competitive advantage [34]. In an ecosystem with significant environmental variability, the first stimulus a seed responds to may not necessarily mark the beginning of a favorable growth season. Therefore, diversifying the timing of germination increases the chances of germinating under suitable environmental conditions [15,35], or germination bet-hedging [36].

The grasses *Eragrostis mexicana* (Hornem.) Lin and *Eragrostis tenuifolia* (A. Rich.) Hochst. ex Steud, Poaceae, inhabit the same microenvironments in Mexico City’s green areas, forming small, single-species or mixed grasslands. Both species have similar growth and reproductive periods. *E. mexicana* is an annual species native to the Southern United States and Argentina. It establishes itself on roads, near cultivated fields, and in disturbed open areas [37]. *E. tenuifolia* is an exotic perennial plant native to Indochina, South Asia, Madagascar, and tropical Africa [38,39]. It is currently distributed in at least eight Mexican states, as well as in South American, European, and Oceanian countries [40]. It establishes itself in environments similar to *E. mexicana*. In addition to being reported as a weed in crops, this species is a potential host of the corn streak virus (MSV), which infects corn and other grasses of economic importance [41,42]. Although this species is not listed on the official list of invasive species in Mexico, two ecologically similar congeneric species that present a very high invasive status do appear as invasive: *E. curvula* [27] and *E. lehmanniana*. They are reportedly capable of displacing native species [43]. Therefore, despite not being considered invasive in Mexico, *E. tenuifolia* is a species with expansion potential whose control could be difficult [44,45]. Against this background, since one of the main goals of invasive species management is identifying potential invaders, trait-related databases must improve the predictive capacity of invasion models that single out highly invasive species [46].

In this study, we experimentally analyzed ecological differences in germination between these two congeneric species and tested whether the proportion, mean time, and synchrony of germination varied among seeds at different ages and under different temperatures. In our experiment, we tested three seed ages (two, three, and four years of storage) and four germination temperatures (15, 20, 25, and 30 °C) under controlled conditions. The comparison between two phylogenetically and ecologically related species provided us with an opportunity to assess the implications of seed banks and the environmental responses to germination on the invasion process of species with broad distribution ranges.

## 2. Results

In all treatments, the germination percentage (GP) of the exotic *E. tenuifolia* was higher than that of the native *E. mexicana*. The generalized linear model (χ^2^ = 88.597, df = 11, *p* < 0.0001) revealed effects of species (χ^2^ = 20.219, df = 1, *p* < 0.001), temperature (χ^2^ = 51.341, df = 3, *p* < 0.001), and the species × seed age interaction (χ^2^ = 17.044, df = 2, *p* = 0.0002). However, species × temperature was not significant (χ^2^ = 2.187, df = 3, *p* = 0.535), suggesting a consistent species-specific response to temperature.

The pilot study suggested that the seeds’ latency period was not imposed by environmental conditions, as <5% of recently harvested seeds for both species germinated under controlled conditions. Seed age had different effects on *E. tenuifolia* and *E. mexicana*. *E. mexicana* seeds tended to germinate more as seed age increased—a trait significantly correlated with vigor and delayed germination, whereas in *E. tenuifolia*, younger seeds had better germination (Figure 1). Germination was similar in *E. mexicana* and *E. tenuifolia* for older seeds (3 and 4 years of storage) but was significantly higher in *E. tenuifolia* for 2-year-old seeds. Rising temperatures had a consistent positive effect on GP for both species; however, at the lowest temperature (15 °C), the GP in *E. mexicana* was close to zero (Figure 1).

Mean germination time (MGT) differed among temperatures (deviation = 2.765, df = 2, *p* < 0.0001), seed ages (deviation = 1.012, gl = 2, *p* < 0.0001), and these factors’ interaction with the species (species × temperature: deviation = 1.01, gl = 2, *p* < 0.0001; species × seed age: deviation = 0.325, gl = 2, *p* < 0.0001). To avoid zero-inflated data that fail to provide adequate MGT assessments, all treatments at 15 °C were excluded because of their very low germination, as well as the replicates without germination in the other treatments (see Figure 2 for the number of replicates per treatment in this analysis). For *E. mexicana*, MGT was significantly lower in seeds with two years of storage compared to the other age groups, while for *E. tenuifolia*, no significant differences among age groups were observed. For temperature, MGT for the native *E. mexicana* was higher in seeds germinated at 30 °C than in the other treatments. In the exotic *E. tenuifolia*, the negative relationship between germination temperature and MGT indicated higher temperatures resulting in accelerated germination (Figure 2).

To calculate synchrony, all treatments at 15 °C were excluded and replicates without germination and replicates with only one germinated seed also were excluded as synchrony could not be calculated (see Figure 3 for the number of replicates per treatment considered in this analysis). This model was only marginally non-significant (χ^2^ = 16.715, df = 9, *p* = 0.053), whereas temperature showed a significant effect (χ^2^ = 7.4063, df = 2, *p* = 0.025), with a tendency for higher synchrony as germination temperature increased (Figure 3).

All relevant data are contained within the manuscript and its Appendix A.

## 3. Discussion

A complex set of biological and physicochemical factors determine the dynamics of a seed bank. Firstly, seed senescence defines the time frame of a seed bank, whether transient (<1 year), short (between 1 and 5 years), or long-lived (>5 years). The response to environmental cues will also influence the transition between the latent and active portions of the seed bank [47]. The interplay between these two factors (the number of viable seeds for germination at any given time, and the number of seeds that germinate) is a tradeoff that ultimately drives the magnitude of the in situ propagule pressure generated by the seed bank of invasive alien species.

The propagule pressure represented by a seed bank increases with extended periods of germination, thereby spreading the success of invasion over time, particularly if a bet-hedging strategy is present (i.e., the idea that some traits reduce short-term reproductive success in favor of longer-term risk reduction [36]). The number of seeds available for germination at any given time partly depends on seed senescence, such that seeds with longer senescence times extend the time frame of propagule pressure. For the two congeneric species, we found that the native species exhibited a positive relationship after years in storage, with older seeds germinating in higher proportions. Once environmental cues triggered the germination of the older seeds, the latent seed bank became an active seed bank. Our experiments indicated that the latency period of native *E. mexicana* seeds slowly decreased with increasing storage time, indicating that a specific environmental condition was required to break latency or trigger germination. These conditions were not fully met when the seeds remained in the field or under experimental conditions. Dry after-ripening periods, although they did not cause an immediate break in latency, gradually reduced it by making the seeds more sensitive to conditions under which they could germinate [48]. By contrast, the exotic *E. tenuifolia* exhibited much higher germination percentages in younger seeds than the native species, and older seeds germinated just as well as those of the native species. This exotic grass appears to require less specific environmental conditions for germination compared to the native *E. mexicana* and can maintain a quiescent seed bank over long periods. Additionally, although the after-ripening period seemed to reduce seed viability, it remained high even after 4 years of storage. Both grass species potentially maintain a seed bank over time (a short-term seed bank), which can offer a constant source of propagules and, thus, a potential source of genetic variability [12]. In exotic species, seed banks such as those found in *E. tenuifolia* positively contribute to naturalization and geographic expansion [8,49], as they enhance the survival and resilience of populations in situations where young or adult individuals might not survive [9].

Germination is one of the routes by which seeds leave the seed bank. The number of seeds that germinate (a component of propagule pressure) partly depends on the number of seeds available for germination and the environmental cues that trigger the germination process. Although the germination rate alone is not considered a definitive trait of invasion success [24,26], evidence suggests that several exotic species have higher germination percentages than their native counterparts [50]. This is the case for grasses such as *Pennisetum setaceum* compared to *Heteropogon contortus* in Hawaii [51], or the invasive *Spartina alterniflora* compared to its native congener *Spartina foliosa* in San Francisco Bay [52]. In Mexico, exotic species with high germination proportions have been reported, which may positively influence their post-introduction success [53,54].

Although exotic species have advantages over native species in germination percentage, their success also depends on specific environmental cues through which germination is triggered [26], and exotics will benefit from a wide response to a variety of environmental cues. In this study, while temperature positively affected both species, the native *E. mexicana* showed very low germination rates at 15 °C, only reaching approximately 50% germination under warmer conditions (30 °C). By contrast, the exotic grass *E. tenuifolia* germinated at all temperatures, with germination rates exceeding 60% from 20 °C onwards. Compared to its native counterpart, exotic *E. tenuifolia*’s ability to germinate at lower temperatures may provide the opportunity for early establishment, as it could begin germinating in response to the temperature rise at the start of the growing season [55]. Although early germination can imply a competitive advantage by enabling early growth and resource acquisition [56], it may also carry the risk of germinating when environmental conditions are not yet ideal for seedling survival [57]. However, *E. tenuifolia* exhibits relatively low germination at 15 °C, along with high germination at higher temperatures. Therefore, its ability to germinate at both low and high temperatures may extend the germination period throughout the growing season, increasing the likelihood of germinating under safe environmental conditions [58]. The broad environmental range in which this species can germinate, coupled with its high germination rates, has positively affected the geographical range in which *E. tenuifolia* has invaded [35]. Although this species is native to Indochina, South Asia, Madagascar, and tropical Africa [38,39], it has also colonized various states in Mexico [59,60,61] and countries across the Americas, Africa, Oceania, and Europe [62] whose environmental conditions widely differ.

Shorter germination time has been linked to the success of exotic species in naturalization [63]. Rapid germination rates can be useful throughout the invasion stages, with generally higher rates in species with greater invasion success [26,63]. However, in this study, we did not find a consistent pattern in the differences in mean germination time between *E. tenuifolia* and *E. mexicana*. The exotic species exhibited a shorter mean germination time than its native counterpart in young seeds, which also happened to germinate in a greater proportion. However, these results were solely for differences observed at 30 °C. At higher temperatures, the mean germination time was lower for the exotic *E. tenuifolia*, while at lower temperatures, the mean germination time was longer than for *E. mexicana*. These differences appear to be due to *E. tenuifolia*’s response to temperature rather than changes in *E. mexicana*’s germination, as the mean germination time of the latter remained constant across different temperatures. Therefore, we can surmise that the mean germination time of the exotic grass *E. tenuifolia* demonstrated a more flexible response to temperature than the native *E. mexicana*.

The exotic *E. tenuifolia* has successfully established itself over a wide geographic range, but it has not been reported as an invasive species, probably because its impact has not yet been quantified. Nevertheless, it is a species that can easily continue to expand [64]. Some of its germination-related traits, such as high seed viability over time, the lack of specific germination cues, the broad range of temperatures at which it can germinate, and its high germination rates, could confer *E. tenuifolia* with advantages during its establishment and naturalization process across a wide geographic range. The presence of a persistent seed bank is especially important in exotic annual grasses. *E. tenuifolia*, on the one hand, may have the advantage of growing earlier in the season (personal observation) and exploiting a niche space before native species respond. It also has high and flexible germination, potentially giving it an edge over slower-responding species.

## 4. Materials and Methods

Seeds were collected from four disturbed areas in the Pedregal de San Angel Ecological Reserve in Mexico City, each ca. half a kilometer apart: one site dominated by *E. mexicana*, one dominated by *E. tenuifolia*, and two where both species coexist. During each of the seed collection events (September 2019, 2020, and 2021), seeds from all four sites were pooled and stored in glass jars under dark conditions. Pilot experiments were conducted with recently collected seeds. The pilot germination experiments consisted of ten replicates of each species with ten seeds each, with a 12 h photoperiod at a constant temperature of 25 ± 3 °C. An experiment was conducted with newly harvested seeds and was repeated every month for 4 months. In all pilot experiments, germination of <5% was obtained for both species.

In November 2021, seeds from all three years of collection were subjected to field conditions by burying the seeds in perforated Eppendorf tubes at a depth of approximately 1.5 cm at two of the collection sites. All seeds were left in these field conditions until May 2022, which marked the start of the first rainy season and the onset of the experiment.

In field conditions, a germination experiment was conducted in Petri dishes with 1% agar under controlled conditions consisting of four constant temperature levels (15, 20, 25, and 30 °C), three different seed ages (two, three, and four years of storage), and two species (*E. tenuifolia* and *E. mexicana*). Each treatment (species × age × temperature) consisted of 10 replicates, with 10 seeds per replicate.

All germination experiments were conducted under a 12 h photoperiod in an environmental chamber (Lumistell ICP-18, Guanajuato, Mexico). Seeds were examined daily for four consecutive days, in addition to a subsequent examination on the seventh day. Germination was considered successful with the visual protrusion of the radicle under a stereoscopic microscope.

Germinability (proportion of germination, GP), mean germination time in days (MGT), and germination synchrony were calculated using the package “GerminaQuant for R” [65].

The synchrony index used by the “GerminaQuant for R” package corresponds to the *Z* index, which was originally proposed to quantify synchrony in flowering and was modified for germination assessment [66]. This index was calculated as follows:$$Z=\sum \frac{C_{n_{i},2}}{N}$$

The mean germination time was calculated as follows:$$MGT=\frac{\sum_{i=1}^{k}n_{i}t_{i}}{\sum_{i=1}^{k}n_{i}}$$
where, $C_{n_{i},2}=\frac{n_{i}(n_{i}-1)}{2}$ represents the number of pairwise combinations of seeds germinated at time *i*; $n_{i}$ is the number of seeds germinated on the ith day; *k* is the last day of germination evaluation; $t_{i}$ is the time from the beginning of the experiment to the ith observation; $N=(\sum n_{i}\left(\sum n_{i-1}\right));and\sum n_{i}$ is the total number of seeds germinated. This index ranges from zero, indicating that each seed germinates at a different time, to one, indicating that all seeds germinate simultaneously.

The effects of seed age and germination temperature on GP were evaluated using a three-way ANOVA with data previously transformed using arcsine. Contrasts between groups were examined with Tukey’s HSD test.

To assess the effects of seed age and germination temperature on synchrony, generalized linear models with a binomial distribution were employed. The independent variables were species, temperature, seed age, and interactions between species and seed age and species and temperature, followed by Tukey’s HSD test.

Due to the lack of observations on days 5 and 6 of the experiment, as well as limited germination on the seventh day of observation, MGT was calculated only considering the first four days of observation. To evaluate the effect of seed age and temperature on MGT, a generalized linear model with a gamma distribution was fitted. The independent variables considered were species, temperature, seed age, and the interactions between species and seed age and species and temperature. Subsequently, contrast tests were conducted using Tukey’s HSD test.

The ANOVA and both GLMs were executed in R version 4.0.0 [67]. Tukey’s HSD tests were performed using the “agricolae” package in the R environment.

## Figures and Tables

**Figure 1 plants-13-00478-f001:**
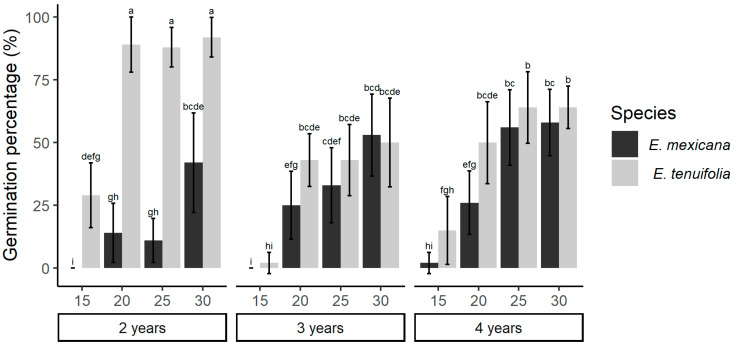
Effects of seed age and temperature on the germination of *E. mexicana* and *E. tenuifolia*. Mean germination percentages (±SD) at four different germination temperatures (15, 20, 25, and 30 °C) of seeds at different ages (two, three, and four years). Different letters indicate significant differences between treatments according to contrast tests (Tukey HSD, *p* ≤ 0.05). In all treatments, *n* = 10.

**Figure 2 plants-13-00478-f002:**
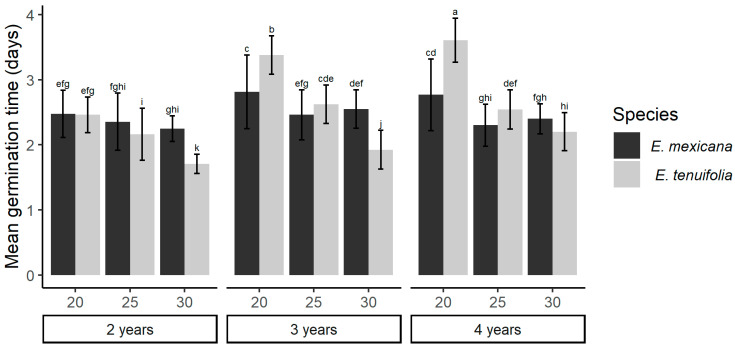
Effects of seed age and temperature on the mean germination times of *E. mexicana* and *E. tenuifolia*. Mean germination times (±SD) at three different germination temperatures (20, 25, and 30 °C) of seeds at different ages (two, three, and four years). Different letters indicate significant differences between treatments according to contrast tests (Tukey HSD, *p* ≤ 0.05). In all *E. tenuifolia* treatments, *n* = 10. *E. mexicana* treatments were performed on two-year-old seeds at 20 °C, *n* = 7, at 25 °C, *n* = 8, and 30 °C, *n* = 9; three-year-old seeds at 20 °C, *n* = 9, at 25 °C, *n* = 9, and 30 °C, *n* = 10; and four-year-old seeds at 20 °C, *n* = 9, at 25 °C *n* = 10, and 30 °C, *n* = 10. Due to no or little germination in treatments at 15 °C, these data were not considered in this analysis.

**Figure 3 plants-13-00478-f003:**
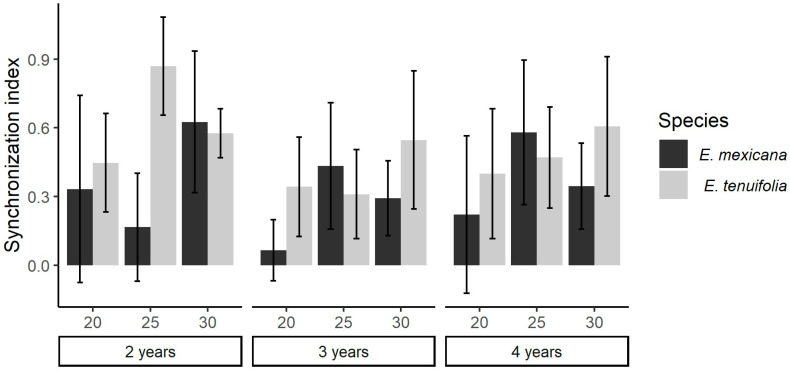
Germination synchrony indexes in *E. mexicana* and *E. tenuifolia* for seeds at different ages and under different temperatures. The synchrony index ranges from zero (seeds germinate on different days) to one (all seeds germinate on the same day). There were no significant differences between treatments. In all *E. tenuifolia* treatments, *n* = 10. *E. mexicana* treatments were performed on two-year-old seeds at 20 °C, *n* = 5, at 25 °C, *n* = 2, and 30 °C, *n* = 9; three-year-old seeds at 20 °C, *n* = 9, at 25 °C, *n* = 9, and 30 °C, *n* = 10; and four-year-old seeds at 20 °C, *n* = 9, at 25 °C, *n* = 10, and 30 °C, *n* = 10.

## Data Availability

All relevant data are within the manuscript and its Appendix A.

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
