# Peer review of "Difference in Germination Traits between Congeneric Native and Exotic Species May Affect Invasion"

_plants, 2024, doi:10.3390/plants13040478_

Round 1

Reviewer 1 Report

Comments and Suggestions for Authors

Seed germination is critical for population establishing and persisting of plant species. It is suggested faster and better germination could facilitate successful invasion of some plants. However, it is not always the case for all invasive plants. It is worthy to explore the germination traits of more invasive species. The authors tested ecological differences in seed germination of exotic Eragrostis Mexicana with a native congeneric species using controlled experiments. The results would help to understand the invasion potential of E. Mexicana. However, the data analysis was not well-designed and described.

1.     t test is not suitable for counted data.

2.     The authors just took one factor into account for each analysis. For example, Figure1(a) just involved seed age, ignoring the different temperature. I don’t think this simple method is good enough to compare the germination traits between native and exotic species.

Comments on the Quality of English Language

The language is well.

Author Response

Estimado crítico 1,

Agradecemos sus comentarios, que enriquecieron significativamente nuestra investigación. A continuación, abordamos punto por punto cada una de sus observaciones:

  1. La prueba t no es adecuada para datos contados.

Se implementaron las siguientes correcciones:

Líneas 308-309: A petición de uno de los revisores, el análisis de proporción de germinación se cambió a un ANOVA con datos transformados por arcoseno. Las pruebas de contraste se realizaron mediante las pruebas de Tukey.

Líneas 313-314: Se realizaron comparaciones post hoc de la sincronía GLM en la germinación utilizando las pruebas de Tukey.

  1. Los autores solo tuvieron en cuenta un factor para cada análisis. Por ejemplo, la Figura 1(a) solo involucra la edad de la semilla, ignorando la diferente temperatura. No creo que este método simple sea lo suficientemente bueno para comparar las características de germinación entre especies nativas y exóticas. Los modelos se realizaron utilizando las variables especie, edad y temperatura y su interacción, como se indica en la primera versión en las líneas 261-265: "Los efectos de la edad de la semilla y la temperatura de germinación sobre el GRP y la sincronía se evaluaron utilizando modelos lineales generalizados con una Distribución binomial. Las variables independientes incluyeron especies, temperatura, edad de las semillas e interacciones entre especies y edades de las semillas, así como entre especies y temperatura […]”. y en las líneas 278-272: "Para evaluar el efecto de la edad y la temperatura de las semillas sobre la MGT, se ajustó un modelo lineal generalizado con una distribución Gamma en R versión 4. 0. 0 [60]. Las variables independientes consideradas incluyeron especie, temperatura, edad de la semilla e interacciones entre especies y edad de la semilla, así como entre especies y temperatura.

El contenido gráfico se presentó por factores, para hacerlos más fáciles de leer. Sin embargo, debido a la falta de claridad, se reemplazaron los tres gráficos para mostrar los resultados de los análisis de manera más completa. Tenga en cuenta que el análisis del porcentaje de germinación que originalmente era un GLM binomial fue reemplazado por un ANOVA de los datos transformados por sugerencia de uno de los revisores; líneas 308-310.

Reviewer 2 Report

Comments and Suggestions for Authors

It is a very innovative idea to study the influence of species invasion ability from the perspective of seed germination in this paper. The author found out the reasons for the invasion of Eragrostis tenuifolia from germination ability with different seed age and germination temperature. Although the research content is very simple, it has certain academic value. It is suggested that the authors supplement the mechanism that germination ability of Eragrostis tenuifolia is superior to that of Eragrostis mexicana. Tell a complete story. Furthermore, some contents in this paper need to be improved, and some results need further analysis and discussion. The comments and suggestions for revision of each part are as follows.

Title: It is suggested to change “favor invasion” to “affect invasion”. The subject is "difference". “Difference” is the difference between two species.

Abstract: It is suggested that the author provide a Graphical Abstract, especially the pictures of the invasion hazards of Eragrostis tenuifolia.

.

1. Introduction: Invasivespecies refers to alien species that have been or will be damaged to economy or environment or endanger human health due to their introduction. The authors do not introduce the invasive hazards of Eragrostis tenuifolia.

2. Results: Suggest adding a subheading to this section.

Line 93:The abbreviation for germination percentage is GP rather than GRP.

Line 93-98:The statistical analysis method is wrong in this part, and data transformation (GP) should be performed first, followed by ANOVA.

Figure 1: Change the ordinate title of Figure 1 to “Germination percentage (%)”. The results of the experiment should be presented as “mean±SD” instead of “mean±SE” with the number of replicates per treatment. The different letters assigned to the means are all wrong in all figures and do not conform to the letter marking method in statistics.

3. Discussion: Suggest adding a subheading to this section.

4. Materials and Methods: Suggest adding a subheading to this section.

Line 250-254: Each treatment consisted of consisted of 10 replicates, with 10 seeds per replicate in this research. Only 10 seeds per replicate is not enough to obtain an accurate germination rate, generally no less than 50 seeds per replicate.

Line 259-260: The calculation formulas of mean germination time in days(MGT) and germination synchrony are not provided. “Germination synchrony index” is “Coefficient of Uniformity of germination, CUG” or “ Coefficient of variation of the germination time, CVt”?

Line 256: “Seeds were examined daily for four days”. This sentence is wrong.

Comments on the Quality of English Language

Minor editing of English language required

Author Response

Dear Reviewer 2,

We appreciate your comments, which significantly enriched our research. Below, we address each of your observations point by point:

Title

  1. It is suggested to change “favor invasion” to “affect invasion”. The subject is "difference". “Difference” is the difference between two species. The title was changed from “Differences in germination traits between a congeneric native and an exotic may favor invasion” to “Difference in germination traits between a congeneric native and an exotic may affect invasion”.

Abstract

  1. It is suggested that the author provide a Graphical Abstract, especially the pictures of the invasion hazards of Eragrostis tenuifolia.

Introduction

  1. The authors do not introduce the invasive hazards of Eragrostis tenuifolia.

The introduction was enriched with information on the ecology of both species. For this purpose, the paragraph " The grasses Eragrostis mexicana (Hornem.) Lin and Eragrostis tenuifolia (A. Rich.) Hochst. ex Steud, Poaceae, inhabit the same microenvironments in Mexico City green areas, forming small, single-species, or mixed grasslands. Eragrostis mexicana is a native annual species that can be found coexisting with the E. tenuifolia that is an exotic perennial. We experimentally analyzed the ecological differences in germination between these two congeneric species, testing whether the proportion, mean time, and synchrony of germination varied among seeds with different ages and under different temperatures. Our experiment tested three seed ages (two, three, and four years of storage) and four germination temperatures (15, 20, 25, and 30° C) under controlled conditions. The comparison between two phylogenetically and ecologically related species provided an opportunity to assess the implications of seed banks and the environmental responses to germination on the success of invasion.", originally located in lines 80-91, was modified to read as follows (lines 80-108):

" The grasses Eragrostis mexicana (Hornem.) Lin and Eragrostis tenuifolia (A. Rich.) Hochst. ex Steud, Poaceae, inhabit the same microenvironments in Mexico City green areas, forming small, single-species or mixed grasslands. Both species have similar growth and reproduction periods. E. mexicana is an annual species native from the southern United States to Argentina. It establishes itself on roads, near cultivated fields and in disturbed open areas [38]. E. tenuifolia is an exotic perennial plant, native to Indochina, South Asia, Madagascar, and tropical Africa [39,40], currently distributed in at least eight states of Mexico, as well as in countries in South America, Europe, and Oceania [41]. It establishes itself in environments similar to E. mexicana. In addition to being reported as a weed in crops, this species is a potential host of the corn streak virus (MSV), which infects corn and other grasses of economic importance [42,43]. Although this species is not listed on the official list of invasive species in Mexico, two ecologically similar congeneric species that present a very high invasive status do appear as invasive, E. curvula [28] and E. lehmanniana, which has been reported as invasive species capable of displacing native species [44]. Therefore, despite not yet being considered invasive in Mexico, E. tenuifolia is a species with expansion potential, and whose control could be difficult [45,46]. As one of the main goals of invasive species management is the identification of potentials invaders there is a need to expand the trait related databases to improve the predictive capacity of invasion models that can be used to single out highly invasive species [47].

We experimentally analyzed the ecological differences in germination between these two congeneric species, testing whether the proportion, mean time, and synchrony of germination varied among seeds with different ages and under different temperatures. Our experiment tested three seed ages (two, three, and four years of storage) and four germination temperatures (15, 20, 25, and 30° C) under controlled conditions. The comparison between two phylogenetically and ecologically related species provided an opportunity to assess the implications of seed banks and the environmental responses to germination on the invasion process of a species with a broad distribution range."

Results

  1. Suggest adding a subheading to this section.

We are not sure what the reviewer wanted here. We do feel the results of the manuscript should not be interpreted individually but as a series of individual results that all contribute to the overall story of how abiotic components and time contribute to advantages in the first stages of the life cycle of invasive and native species.

  1. Line 93: The abbreviation for “germination percentage” is GP rather than GRP. Changes made to lines 111, 124, 125, 292, 308.
  2. Line 39-98: The statistical analysis method is wrong in this part, and data transformation(GP) should be performed first, followed by ANOVA. It was done, lines 308-310. The binomial GLM was changed by two-way ANOVA, using the species, seed age, and temperature as explanatory factors. To do so, the germination percentage data were transformed by arcsine. We still think the GLM is more appropriate though as it considers the binomial nature of the response variable.
  3. Figure1: Change the ordinate title of Figure 1 to “Germination percentage (%)”. The results of the experiment should be presented as “mean ±SD” instead of “mean±SE” with the number of replicates per treatment. The different letters assigned to the means are all wrong in all figures and do not conform to the letter marking method in statistics. Comments addressed in three graphs:
  • The ordinate title of Figure 1 was changed to “Germination percentage (%)”.
  • The results are now displayed as the mean±SE.
  • The arrangement of letter marks according to the contrast test (TukeyHSD) was corrected.

Discussion

  1. Suggest adding a subheading to this section.

We are not sure what the reviewer wanted here. We do feel the results of the manuscript should not be interpreted individually but as a series of individual results that all contribute to the overall story of how abiotic components and time contribute to advantages in the first stages of the life cycle of invasive and native species.

Materials and method

  1. Suggest adding a subheading to this section. We think that adding subsections to a paper that is not overly long would atomize the information. We have tried to integrate the information into a single story more than isolated topics.
  2. Line 250-254: Each treatment consisted of consisted of 10 replicates, with 10 seeds per replicate in this research. Only 10 seeds per replicate is not enough to obtain an accurate germination rate, generally no less than 50 seeds per replicate. The accuracy of germination will depend on how well seeds germination and their variation. There is no rule of thumb as it will depend on seed behavior. Oliviera et al 2016 (Olivieria et al, 2016. Australian J. Bot 64 (4)) showed that for seeds that germinate well, sample sizes below 100 seeds were suitable. For out study, 100 seeds per treatment were used for each combination treatment.
  3. Line 259-260: The calculation formulas of mean germination time in days (MGT) and germination synchrony are not provided. Germination synchrony index” is “Coefficient of Uniformity of germination, CUG” or “ Coefficient of variation of the germination time, CVt”?

The synchrony index used was the Z index, which was originally proposed to quantify synchrony in flowering; but it was modified to measure it in germination [67]. This, and the formula of mean germination time in days are now explained in the lines:

294-306: The synchrony index used by the "GerminaQuant for R" package corresponds to the Z index, that was originally proposed to quantify synchrony in flowering; but modified for germination assessment [67]. This index was calculated as follows:

[See formula in line 27 of the new manuscript]

The mean germination time as:

[See formula in line 300 of the new manuscript]

Where [formula], representing the combination of the seeds germinated in the time i, two together.  is the number of seeds germinated in the time [Formula]; k is the last day of germination evaluation; ti is the time from the beginning of the experiment to the  observation; [formula]; and Sni is the total number of seeds germinated. This index ranges from zero, indicating each seed germinates on a different time, to one, indicating all seeds germinate simultaneously.

  1. Line 256: “Seeds were examined daily for four days”. This sentence is wrong. The sentence was changed to “Seeds were examined on a daily basis for a period of four consecutive days”.

Reviewer 3 Report

Comments and Suggestions for Authors

The article is devoted to an actual problem and contains a relevant data for the readers. English could be improved as well as there is not enough data in introduction. Introduction must be improved and new literature data could be added. The conclusions must be written and supported by data.

Comments on the Quality of English Language

Inglish must be improved

Author Response

Dear Reviewer 3,

We appreciate your comments, which significantly enriched our research. Below, we address each of your observations point by point.

  1. English could be improved as well as there is not enough data in introduction.
  2. Introduction must be improved and new literature data could be added.

The introduction was enriched with information on the ecology of both species. For this purpose, the paragraph " The grasses Eragrostis mexicana (Hornem.) Lin and Eragrostis tenuifolia (A. Rich.) Hochst. ex Steud, Poaceae, inhabit the same microenvironments in Mexico City green areas, forming small, single-species, or mixed grasslands. Eragrostis mexicana is a native annual species that can be found coexisting with the E. tenuifolia that is an exotic perennial. We experimentally analyzed the ecological differences in germination between these two congeneric species, testing whether the proportion, mean time, and synchrony of germination varied among seeds with different ages and under different temperatures. Our experiment tested three seed ages (two, three, and four years of storage) and four germination temperatures (15, 20, 25, and 30° C) under controlled conditions. The comparison between two phylogenetically and ecologically related species provided an opportunity to assess the implications of seed banks and the environmental responses to germination on the success of invasion.", originally located in lines 80-91, was modified to read as follows (lines 80-108):

" The grasses Eragrostis mexicana (Hornem.) Lin and Eragrostis tenuifolia (A. Rich.) Hochst. ex Steud, Poaceae, inhabit the same microenvironments in Mexico City green areas of, forming small, single-species or mixed grasslands. Both species have similar growth and reproduction periods. E. mexicana is an annual species native from the southern United States to Argentina. It establishes itself on roads, near cultivated fields and in disturbed open areas [38]. E. tenuifolia is an exotic perennial plant, native to Indochina, South Asia, Madagascar, and tropical Africa [39,40], currently distributed in at least eight states of Mexico, as well as in countries in South America, Europe, and Oceania [41]. It establishes itself in environments similar to E. mexicana. In addition to being reported as a weed in crops, this species is a potential host of the corn streak virus (MSV), which infects corn and other grasses of economic importance [42,43]. Although this species is not listed on the official list of invasive species in Mexico, two ecologically similar congeneric species that present a very high invasive status do appear as invasive, E. curvula [28] and E. lehmanniana, which has been reported as invasive species capable of displacing native species [44]. Therefore, despite not yet being considered invasive in Mexico, E. tenuifolia is a species with expansion potential, and whose control could be difficult [45,46]. As one of the main goals of invasive species management is the identification of potentials invaders there is a need to expand the trait related databases to improve the predictive capacity of invasion models that can be used to single out highly invasive species [47].

We experimentally analyzed the ecological differences in germination between these two congeneric species, testing whether the proportion, mean time, and synchrony of germination varied among seeds with different ages and under different temperatures. Our experiment tested three seed ages (two, three, and four years of storage) and four germination temperatures (15, 20, 25, and 30° C) under controlled conditions. The comparison between two phylogenetically and ecologically related species provided an opportunity to assess the implications of seed banks and the environmental responses to germination on the invasion process of a species with a broad distribution range."

With this, bibliographical sources were also added.

  • Bittencourt H von H, Bonome LT da S, Pagnoncelli Junior F de B, Lana MA, Trezzi MM. Seed germination and emergence of Eragrostis tenuifolia (A. Rich.) Hochst. ex Steud. in response to environmental factors. J Plant Prot Res [Internet]. 2016;56(1):32–8. Available from: https://doi.org/10.1515/jppr-2016-0005
  • Calderón G, Rzedowski J. Flora fanerogámica del Valle de México. Instituto de Ecología, A.C. 2001.
  • Flora of North America North of Mexico(). Eragrostis mexicana. Available from: http://floranorthamerica.org/Eragrostis_mexicana Retrieved 6 December 2023.
  • Fourier A, Penone C, Pennino MG Courchamp F. 2019. Predicting future invaders and future invasions. PNAS 116: 7905-910. https://doi.org/10.1073/pnas.1803456116
  • Goulart ICGR, Merotto Junior A, Perez NB, Kalsing A. Controle de capim-annoni-2 (Eragrostis plana) com herbicidas pré-emergentes em associação com diferentes métodos de manejo do campo nativo. 27, Planta Daninha. scielo ; 2009.
  • Jung MJ, Veldkamp JF, Kuoh CS. Notes on Eragrostis wolf (Poaceae) for the flora of Taiwan. Taiwania. 2008.
  • Njuguna, J. G. M., Gordon, D. T., & Louie, R. (1997). Wild grass hosts of maize streak virus and its Cicadulina leafhopper vectors in Kenya. In 5. Proceedings of the Eastern and Southern Africa Regional Maize Conference, Arusha (Tanzania), 3-7 Jun 1996. CIMMYT.
  • Peterson PM, Vega IS. Eragrostis (Poaceae: Chloridoideae: Eragostidae: Eragrostidinae) of Peru. Ann Missouri Bot Gard [Internet]. 2007 Dec 1;94(4):745–90. Available from: https://doi.org/10.3417/0026-6493(2007)94[745:EPCEEO]2.0.CO
  • POWO (2023). "Plants of the World Online. Facilitated by the Royal Botanic Gardens, Kew. Available from: http://www.plantsoftheworldonline.org/ taxon/urn:lsid:ipni.org:names:1064189-2#other-data Retrieved 11 December 2023.
  • Sanchez-Muñoz A de J. Invasive lehmann lovegrass (Eragrostis lehmanniana) in Chihuahua, Mexico: Consequences of invasion. Oklahoma State University; 2009.
  1. The conclusions must be written and supported by data. We think the discussion is now centered on the results we obtained. We also consider the importance of the seed bank in a management scenario.

Reviewer 4 Report

Comments and Suggestions for Authors

Dear Editor and authors,

The research topic is interesting and important for understanding some of the mechanisms behind the success of invasive species.

I have some suggestions for the improvement of the manuscript. They are written as comments in the file with the manuscript, and here are more general comments:

1. Introduction: Please write more about the areals and ecology of both species, as the information is essential for the interpretation of the results

2. Methods:

- There is no information about the pilot experiment.
- Explain what contrast tests are.

3. Results: Please structure the results to be more explicit and understandable:

- It is not clear what data are included in graphs. Please write more clearly. For example, in graphs about the seed age, it is unclear if you included the experiments at all temperatures or only selected. Which one, in this case?

- In some of the graphs, the marks for the statistic tests are missing.

- Explain more precisely about the excluded data

4. Discussion: Do you have any information about the survival and further development of the seedlings?

With best wishes!

Author Response

Dear Reviewer 4,

We appreciate your comments, which significantly enriched our research. Below, we address each of your observations point by point.

Introduction

  1. Please write more about the areas and ecology of both species, as the information is essential for the interpretation of the results.

The introduction was enriched with information on the ecology of both species. For this purpose, the paragraph " The grasses Eragrostis mexicana (Hornem.) Lin and Eragrostis tenuifolia (A. Rich.) Hochst. ex Steud, Poaceae, inhabit the same microenvironments in Mexico City green areas, forming small, single-species, or mixed grasslands. Eragrostis mexicana is a native annual species that can be found coexisting with the E. tenuifolia that is an exotic perennial. We experimentally analyzed the ecological differences in germination between these two congeneric species, testing whether the proportion, mean time, and synchrony of germination varied among seeds with different ages and under different temperatures. Our experiment tested three seed ages (two, three, and four years of storage) and four germination temperatures (15, 20, 25, and 30° C) under controlled conditions. The comparison between two phylogenetically and ecologically related species provided an opportunity to assess the implications of seed banks and the environmental responses to germination on the success of invasion.", originally located in lines 80-91, was modified to read as follows (lines 80-108):

" The grasses Eragrostis mexicana (Hornem.) Lin and Eragrostis tenuifolia (A. Rich.) Hochst. ex Steud, Poaceae, inhabit the same microenvironments in Mexico City green areas of, forming small, single-species or mixed grasslands. Both species have similar growth and reproduction periods. E. mexicana is an annual species native from the southern United States to Argentina. It establishes itself on roads, near cultivated fields and in disturbed open areas [38]. E. tenuifolia is an exotic perennial plant, native to Indochina, South Asia, Madagascar, and tropical Africa [39,40], currently distributed in at least eight states of Mexico, as well as in countries in South America, Europe, and Oceania [41]. It establishes itself in environments similar to E. mexicana. In addition to being reported as a weed in crops, this species is a potential host of the corn streak virus (MSV), which infects corn and other grasses of economic importance [42,43]. Although this species is not listed on the official list of invasive species in Mexico, two ecologically similar congeneric species that present a very high invasive status do appear as invasive, E. curvula [28] and E. lehmanniana, which has been reported as invasive species capable of displacing native species [44]. Therefore, despite not yet being considered invasive in Mexico, E. tenuifolia is a species with expansion potential, and whose control could be difficult [45,46]. As one of the main goals of invasive species management is the identification of potentials invaders there is a need to expand the trait related databases to improve the predictive capacity of invasion models that can be used to single out highly invasive species [47].

We experimentally analyzed the ecological differences in germination between these two congeneric species, testing whether the proportion, mean time, and synchrony of germination varied among seeds with different ages and under different temperatures. Our experiment tested three seed ages (two, three, and four years of storage) and four germination temperatures (15, 20, 25, and 30° C) under controlled conditions. The comparison between two phylogenetically and ecologically related species provided an opportunity to assess the implications of seed banks and the environmental responses to germination on the invasion process of a species with a broad distribution range."

Materials and methods

  1. There is no information about the pilot experiment.

The sentence “. Pilot experiments were carried out with recently collected seeds. The pilot germination experiments consisted of ten replicates of each species with 10 seeds each, with a 12-hour photoperiod at a constant temperature of 25±3°C. An experiment was carried out with newly harvested seeds, and it was repeated every month for 4 months. In all pilot experiments, germination of less than 5% were obtained for both species.” In the lines 272-277.

  1. Explain what contrast tests are. All contrast tests were performed with Tukey tests. Now it is specified in:

Lines 307-309: “The effects of seed age and germination temperature on GP were evaluated using a two-way ANOVA with data previously transformed by arcsine. Contrasts between groups were performed with Tukey's HSD test.”

Lines 311-313: “The independent variables included species, temperature, seed age, and interactions between species and seed age, as well as between species and temperature followed by Tukey's HSD test.”

Lines 317-320: The independent variables considered included species, temperature, seed age, and inter-actions between species and seed age, as well as between species and temperature. Subsequently, contrast tests were conducted using Tukey's HSD test.

Results

  1. It is not clear what data are included in graphs. Please write more clearly. For example, in graphs about the seed age, it is unclear if you included the experiments at all temperatures or only selected. Which one, in this case? The graphics was presented by factor, to make them more reader friendly. But, in all analysis and in all graphs the three explanatory variables were considered (species, seed age, and temperature), and their interaction. However, because the graphs turned out to be less clear than expected, the three graphs were replaced to show the results of the analyzes more fully, that is, the results of each treatment without grouping.

The n of each treatment was placed in each graph.

Note that the germination percentage analysis that was originally a binomial GLM was replaced by an ANOVA of the transformed data at the suggestion of one of the reviewers; lines 307-309.

  1. In some of the graphs, the marks for the statistic tests are missing. The arrangement of letter marks according to the contrast test (TukeyHSD) were corrected in graphs 1-2. The letter marks were deleted in graph 3 because the model of synchrony is only marginally non-significant.

  1. Explain more precisely about the excluded data.
  2. The sentence “Treatments at 15° C with very low germination and replicates with no germination were excluded.” was changed to “In order not have zero inflated data that do not provide adequate MGT assessments, all treatments at 15°C were excluded because they of very low germination, as well as the replicates without germination of the other treatments (see Figure 2 for the number of replicates per treatment considered in this analysis).” in the lines 137-141.

And the sentence “For the calculation of synchrony, all treatments at 15°, replicates without germination were also excluded and the replicates with only one germinated seed as synchrony cannot be calculated (see Figure 3 for the number of replicates per treatment considered in this analysis).” was added in the lines 157-161.

Discussion

  1. Do you have any information about the survival and further development of the seedlings?

No, we did not follow the fate of the seedlings beyond the 4 day period.

In file comments

  1. Line 17: (add “in Mexico”): Line 15-18. The sentence “We characterized 15 seed germination traits and how they influence the success of Eragrostis mexicana, a native species, 16 and Eragrostis tenuifolia, an exotic species (Poaceae) in the context of their potential for biological 17 invasion.” was changed to “We characterized seed germination traits and how they influence the success of Eragrostis mexicana, a native species, and Eragrostis tenuifolia, an exotic species (Poaceae) in Mexico, in the context of their potential for biological invasion.”

  1. Line 19: Linea 18 – 20. The sentence “Seeds from both species were collected from four sites in a natural protected site, in Mexico City, and germination experiments were conducted under different temperatures and seed ages.” was changed to “Seeds from both species were collected from four sites in a natural protected site, in Mexico City, and germination of seeds of different seed ages experiments were conducted under different temperatures.”

  1. Line 83: please, describe, the original areal of the species and also the area where the species in invasive. Please, shortly describe the ecology of both species as this is important for the interpretation of the results. Answer in comment 1 of the introduction.

  1. Line 99: it is not clear, what is the ripening period. In the lines 117-119 we have added the following text “The pilot study suggested that seeds had a latency period not imposed by environmental conditions, as less than 5% of recently harvested seeds of both species germinated under controlled conditions”

  1. Line 103: it is not clear, what are the T conditions for this experiment.

The models were carried out using the variables species, age, and temperature and their interaction. But the graphs are presented by factor, to make them more reader friendly. However due to the lack of clarity, the three graphs were replaced to show the results of the analyzes more fully. Note that the germination percentage analysis that was originally a binomial GLM was replaced by an ANOVA of the transformed data at the suggestion of one of the reviewers; lines 307-309.

  1. Line 107: Figure 1B, please, mark the results of statistical analysis in the graph (same as on the left): All graphs were modified and all the marks of the statistical results are now shown.

  1. Line 118: how many of the replicates had no germination, at what treatments?

The sentence “Treatments at 15° C with very low germination and replicates with no germination were excluded.” was changed to “In order not have zero inflated data that do not provide adequate MGT assessments, all treatments at 15°C were excluded because they of very low germination, as well as the replicates without germination of the other treatments (see Figure 2 for the number of replicates per treatment considered in this analysis).” in the lines 137-141.

And the sentence “For the calculation of synchrony, all treatments at 15°, replicates without germination were also excluded. and the replicates with only one germinated seed (see Figure 3 for the number of replicates per treatment considered in this analysis).” Was added in the lines 155-157.

  1. Line 127: I have the same questions as for graphs on figure 1. All graphs were modified and all the marks of the statistical results are now shown.

  1. Line 138: in Figure 3a mark the results of statistical tests. All graphs were modified and all the marks of the statistical results are now shown.

  1. Line 243: you shoud decsribe the conditions and methods of pilot studies too.

This is now addressed in the answer to question 1 of Materials and methods.

  1. Line 272: please, describe, what kind of tests are included in this statistics.
  2. The sentences “To evaluate the effect of seed age and temperature on MGT, a generalized linear model with a Gamma distribution was fit in R version 4. 0. 0 [60]. The independent variables considered included species, temperature, seed age, and interactions between species and seed age, as well as between species and temperature. Subsequently, contrast tests were conducted using the emmeans in R [61].” were changed to” To evaluate the effect of seed age and temperature on MGT, a generalized linear model with a Gamma distribution was fitted. The independent variables considered included species, temperature, seed age, and interactions between species and seed age, as well as between species and temperature. Subsequently, contrast tests were conducted using Tukey's HSD test.”. Lines 316-320.

Round 2

Reviewer 1 Report

Comments and Suggestions for Authors

No more comments now.

Reviewer 2 Report

Comments and Suggestions for Authors

The problems have been well corrected and it is recommended that the paper be accepted.